# Influences of CO$_2$ on the Microstructure in Sheared Olivine  Aggregates

**Huihui Zhang** [1,2,3], **Ningli Zhao** [4], **Chao Qi** [1,2], **Xiaoge Huang** [1,2,*] and **Greg Hirth** [4]

1   Key Laboratory of Earth and Planetary Physics, Institute of Geology and Geophysics, Chinese Academy of Sciences, Beijing 100029, China; zhanghuihui@mail.iggcas.ac.cn (H.Z.); qichao@mail.iggcas.ac.cn (C.Q.)
2   Innovation Academy for Earth Science, Chinese Academy of Sciences, Beijing 100029, China
3   University of Chinese Academy of Sciences, Beijing 100049, China
4   Department of Earth, Environmental and Planetary Sciences, Brown University, Providence, RI 02906, USA; ningli_zhao@brown.edu (N.Z.); greg_hirth@brown.edu (G.H.)
*   Correspondence: xghuang@mail.iggcas.ac.cn; Tel.: +86-010-82998429

**Abstract:** Shear deformation of a solid-fluid, two-phase material induces a fluid segregation process that produces fluid-enriched bands and fluid-depleted regions, and a crystallographic preferred orientation (CPO) characterized by girdles of [100] and [001] axes sub-parallel to the shear plane and a cluster of [010] axes sub-normal to the shear plane, namely the AG-type fabric. Based on experiments of two-phase aggregates of olivine + basalt, a two-phase flow theory and a CPO formation model were established to explain these microstructures.  Here, we investigate the microstructure in a two-phase aggregate with supercritical CO$_2$ as the fluid phase and examine the theory and model, to evaluate differences in rheological properties due to the presence of CO$_2$ or basaltic melt. We conducted high-temperature and high-pressure shear deformed experiments at 1 GPa and 1100 °C in a Griggs-type apparatus on samples made of olivine + dolomite, which decomposed into carbonate melt and CO$_2$ at experimental conditions.  After deformation, CO$_2$ segregation and an AG-type fabric were observed in these CO$_2$-bearing samples, similar to basaltic melt-bearing samples.  An SPO-induce CPO model was used to explain to the formation of the fabric. Our results suggest that the influences of CO$_2$ as a fluid phase on the microstructure of a two-phase olivine aggregate is similar to that of basaltic melt and can be explained by the CPO formation model for the solid-fluid  system.

**Keywords:** olivine aggregates; CO$_2$; crystallographic preferred orientation; AG-type fabric

## 1. Introduction

In the mantle of the Earth, deformation often occurs in regions where melt is produced and transported, such as plate boundaries, plumes, intra-plate rifts, and boundary layers. Deformation and melt transport, including melt migration and segregation, are strongly coupled in partially molten rocks, as demonstrated by field observations [1], laboratory experiments [2], and theoretical analyses [3]. Significantly, melt spontaneously localizes into melt-enriched bands oriented ∼20° to the shear plane, antithetic to the shear direction in partially molten rocks deformed in general and torsional shear [4–7]. The formation of melt-enriched bands is predicted and modeled by the two-phase flow theory with anisotropic viscosity [3,8,9].  Moreover, shear deformation of partially molten, olivine aggregates produces a crystallographic preferred orientation (CPO), which is characterized by a cluster of [010] axes sub-normal to the shear plane and girdles of [100] and [001] axes sub-parallel to the shear plane [2,10]. This CPO, often referred to as AG-type fabric [11], can form due to the development of a shape preferred orientation (SPO), where the grain shape is crystallographically controlled in partially molten olivine aggregates [10].

Deformation-induced microstructural features in partially molten rocks, primarily the re-distribution of melt and the formation of CPOs as introduced above, have mostly been investigated in the solid-fluid system consisting of olivine and basaltic melt [12–14]

or other silicate melt [7]. These fluid phases have low dihedral angles, 30–50° [15–17], relatively high viscosities (>100 Pa·s for basaltic melt at 1473 K [7]), and high solubility for the solid phase [18,19]. These properties of the fluid phases were incorporated in the development of previous theory for melt distribution and mechanism for CPO formation. However, whether the theory and mechanism apply to fluid phases with very different properties is yet to be examined. Supercritical carbon dioxide ($CO_2$), a fluid state of carbon dioxide at or above its critical temperature and pressure, is a fluid phase suitable for the examination. A laboratory study found dihedral angles for olivine + $CO_2$-rich fluid of ~80–90° [20]. By constrast, the dihedral angle of olivine + carbonate melt is much lower (~30°) [21]. A model extrapolatd from laboratory measurements suggests that the viscosity of $CO_2$ is lower than 0.1 mPa·s at 1 GPa and 900 K [22]. Conductivity experiments carried out on peridotite showed that adding carbonate melt significantly increases the electrical conductivity [23], indicating that the presence of carbonate melt enhances the transport properties. Moreover, $CO_2$ and carbonate melt are also important for the dynamics of the mantle.

With the mantle being the largest carbon reservoir, carbon cycles between the mantle and the atmosphere via volcanic activity and subduction [24], where intense deformation occurs. In the upper mantle, carbon exists as $CO_2$ or carbonate melt [25]. At certain geological conditions, carbonate melt will decompose into $CO_2$, which can be held stable in the lithosphere [26]. Thus, the influence of $CO_2$ and carbonate melt on the microstructures in olivine aggregates not only aids the understanding of the microstructural evolution of partially molten rocks, but also benefits the study of carbon transportation in the mantle.

In the present paper, we present results from laboratory deformation experiments on olivine + $CO_2$ aggregates at high temperature and pressure. Detailed microstructural analyses of sheared samples provide observations on the distribution of $CO_2$ and the development of CPOs. In this contribution, we focus on the CPOs. A comparison between CPOs observed in this study and those reported in olivine + basalt samples reveal the mechanism for CPO formation in this solid-fluid, two-phase system.

## 2. Experimental Methods

### 2.1. Sample Preparation

Sample powders contained 88 wt% olivine, 3 wt% clinopyroxene (CPx, with ~8 wt% of orthopyroxene; [27]) and 9 wt% dolomite. Olivine and CPx crystals were hand-picked from a lherzolite xenolith (Damaping, Hebei, China; [28]). The composition of the dolomite is $CaMg(CO_3)_2$. Olivine, CPx, and dolomite were separately ground in an agate motor and sieved to a particle size of <32 μm. Powders of olivine and CPx were subsequently baked in a furnace for 20 h at 850 °C and an oxygen fugacity set by flowing CO and $CO_2$ at a ratio of 1:1, which is within the stability field of natural olivine ($f_{O2} = 10^{-14}$ Pa). Then powders were mechanically mixed.

Each sample was prepared by cold-pressing ~0.1 g of the mixture into a nickel can between two alumina pistons beveled at 45°. The details of the experimental assembly are illustrated in Figure 1a. Before hot-pressing, cold-pressed samples were stored in a vacuum oven at 110 °C for more than 6 h. Cold-pressed samples were then hot-pressed at 1100 °C and a confining pressure of 1 GPa for 17–18 h in a Griggs-type apparatus.

### 2.2. Deformation Experiments

Immediately after hot-pressing, samples were deformed at 1100 °C and 1 GPa. Two experiments (W2264 and W2266) were carried out at a constant axial displacement rate of $7.5 \times 10^{-5}$ mm/s to a shear strain of $\gamma \approx 3$. After deformation, one experiment (W2266) was then annealed for 10 h at 1100 °C and 1 GPa. A strain-rate-stepping experiment (W2263) was carried out at axial displacement rates of $7.5 \times 10^{-5}$ mm/s, $1.9 \times 10^{-5}$ mm/s, $1.9 \times 10^{-4}$ mm/s, and $7.5 \times 10^{-5}$ mm/s to a total shear strain of $\gamma \approx 5$.

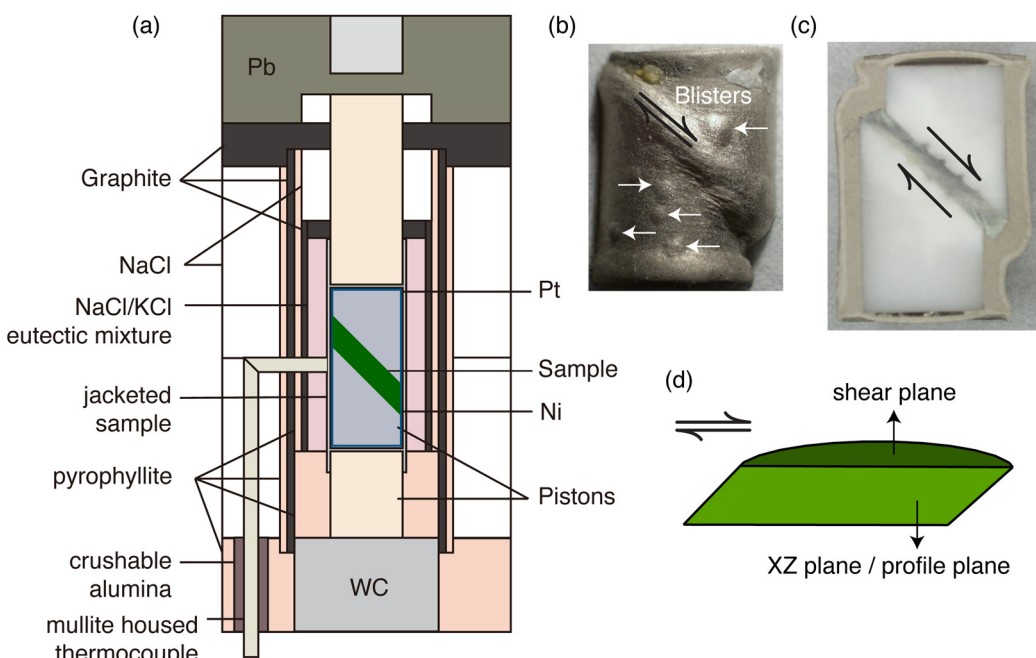

**Figure 1.** (**a**) A sketch of the experimental assembly. (**b**) A photo of a deformed sample. Blisters on the surface of the Ni jacket are marked by white arrows. (**c**) A photo of a sample cut through the profile plane. (**d**) Schematic illustration of the shear plane and profile plane.

At the end of each experiment, the sample was first quenched to 800 °C and then the temperature was decreased to 300 °C in less than 10 min. The confining pressure was decreased while maintaining a differential stress of less than 100 MPa at 300 °C to minimize unloading cracks. When the confining pressure decreased to 500 MPa, we started to decrease the temperature again. With the confining pressure decreasing to 0 MPa, the temperature was gradually lowered to the temperature of the cooling water (19 °C).

### 2.3. Mechanical Data Processing

Axial displacement, load, temperature and confining pressure data were collected every second in our apparatus. The axial displacement rate was multiplied by $\sqrt{2}$ and divided by the measured thickness of samples, to obtain the shear strain rate. Equivalent strain rate was obtained by dividing the shear strain rate by $\sqrt{3}$. The thinning of specimens resulted in a small increase of shear strain rate with an imposed constant axial displacement rate. We obtained a linear correlation between the specimen thickness and shear strain, which was incorporated into the calculation of shear strain rate. Comparison of hot-pressed and deformed specimens indicates that the sample thickness decreased from ~1.3 mm to ~0.9 mm after a shear strain of $\gamma = 3$, which increased the shear strain rate by 40% (Appendix A, Figure A1).

Axial load was converted to axial differential stress by dividing the axial load by the area of the horizontal cross section of the pistons. Axial differential stresses were corrected for force contributions from assembly friction by subtracting the friction estimated by an intercept of two linear fits of force-displacement curves just before and just after the specimen "hit point" [29]. Shear stress was calculated by dividing the differential stress by 2. Equivalent stress was then obtained by applying a factor of $\sqrt{3}$ to the shear stress. The accuracy of stress data is estimated to be $\pm 5$ MPa.

### 2.4. Microstructure Analysis

After deformation, the nickel jacket was peeled off, and the alumina pistons were cut off. Each sample was cut perpendicular to the shear plane (referred to as "profile plane", as illustrated in Figure 1d). The sample section was dry-polished on a series of diamond

lapping films down to 0.5 μm. To protect the deformation microstructures of the highly porous samples, water, ethanol or other fluids were not used during mechanical polishing. After dry polishing, the sample was polished with 0.02 μm $SiO_2$ non-crystallizing colloidal silica for ~4 h. In two samples (W2263 and W2264), a section was also cut parallel to the shear plane (Figure 1d) and polished in the same manner as described above.

Polished sections were prepared for analysis by electron backscatter diffraction (EBSD). To obtain crystallographic orientation data, high-resolution orientation maps with a step size of 0.5 μm were collected for all sections using a Zeiss Sigma field emission SEM equipped with an Oxford Instruments Nordlys Nano EBSD detector (State Key Laboratory of Earthquake Dynamics, Institute of geology China Earthquake Administration, Beijing, China) and a Tescan MIRA3 LMU field emission scanning electron microscope equipped with an Oxford Instruments NordlysMax2 EBSD detector (Department of Earth and Environmental Sciences, Boston College, Boston, USA) (only for W2264). Raw orientation data were processed with HKL Channel5 software, including removal of single misindexed points, assigning unindexed points the average orientation of neighboring grains, and removal of systematic misindexed points, as outlined in previous studies [30,31].

Grains were constructed from processed orientation data using the MTEX toolbox [32] in MATLAB. Grain boundaries were drawn where neighboring pixel misorientations exceeded 10°. No extrapolation of orientation data was applied in MTEX, because the data were already processed by the HKL Channel5 software. Grain size was determined by applying a factor of $4/\pi$ [33] to the equivalent diameter of a circle with the area of each grain in cross section. In the analysis of the average grain size for a map, grains containing less than four pixels or lying on the edge of the map were excluded.

To analyze the grain shape, each grain was fit with an ellipse to obtain its aspect ratio and the orientation of the long axis using MTEX toolbox [32]. Grains with an area of less than 2 μm$^2$ were excluded, because they did not contain enough pixels to derive reliable values of aspect ratio and slope of the ellipse. Then, the shape preferred orientation (SPO) was illustrated by a rose diagram, and the distribution of aspect ratios was illustrated by a histogram. An analysis of the SPOs in grains grouped with respect to their crystallographic orientations was performed following the methods described in Qi et al. [10] in the profile and shear planes.

Orientation distributions were generated from the mean orientations of at least 500 grains with a half-width angle of 10° using the MTEX toolbox in MATLAB [34,35]. To facilitate comparison with previous studies, orientation data were rotated so that the sense of shear is top to the right. To quantify the strength of the CPOs, both the J-index [36] and the M-index [37] were used.

After EBSD analysis, the sample was etched in phosphoric acid for 45 min to highlight grain boundaries in preparation for analysis by scanning electron microscopy (SEM). After coating the sections with >8 nm thick carbon, backscattered electron images were collected using a Zeiss Sigma field emission scanning electron microscope at an accelerating voltage of 15 kV. The $CO_2$ + melt fractions for each sample were estimated in SEM images by assuming these phases occupied all the pockets.

*2.5. Composition Analysis*

Electron microprobe analyses (EPMA) were performed on a SXFive. Samples were analyzed using an accelerating voltage of 15 kV, a beam current of 20 nA and a focused beam. The $CO_2$ in the melts may be estimated by difference between 100% and the observed microprobe totals [38].

**3. Results**

*3.1. Starting Materials*

In our samples, 9 wt% dolomite was added to the olivine aggregates as a $CO_2$ source. At our experimental conditions (1100 °C and 1 GPa), dolomite thermally decomposes and melts, producing supercritical $CO_2$ and carbonate melt [39,40]. After samples (both

hot-pressed and deformed) were removed from the apparatus, many "blisters" started to appear on the surface of the Ni jacket (Figure 1b). Piercing through the Ni jacket with the sample submerged in ethanol, gas bubbles were released from the blisters for several minutes—indicating $CO_2$ was successfully encapsuled in the assembly by the jacket at our experimental conditions.

The presence of $CO_2$ was also demonstrated by the microstructures presented in Figure 2a,b. Voids (Figure 2b) located at grain boundaries and junctions are interpreted as $CO_2$ fluid, in accordance with the results of [41]. Carbonate melt was homogeneously distributed in the hot-pressed sample. Most $CO_2$ and melt pockets were located at three- and four-grain junctions and grain boundaries of the solid phase, with no melt-preferred orientation observed. Individual pockets appeared to be made of either the melt phase or the subcritical $CO_2$. Owing to the delicate nature of these pore structures, we cannot rule out that some of the larger pores were "pluck-outs". By image analysis, the porosity in the hot-pressed sample was approximately 7%, including the $CO_2$, melt, and pluck-outs.

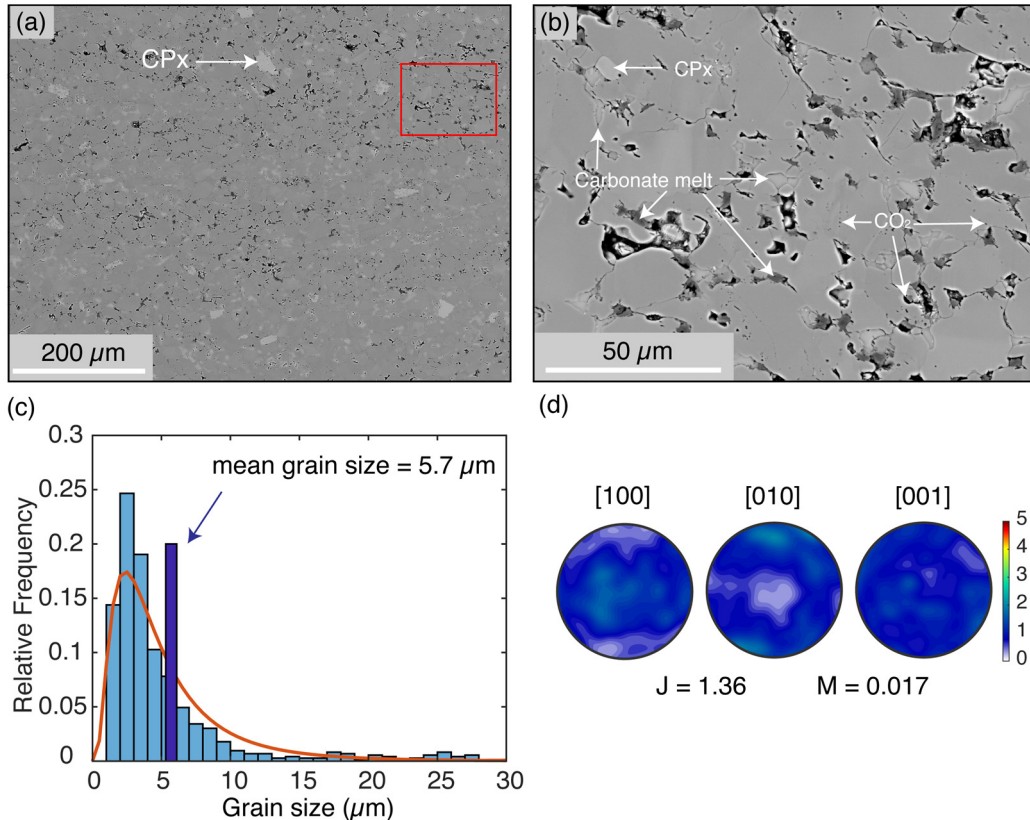

**Figure 2.** The backscattered electron (BSE), grain size, and crystallographic orientations of the hot-pressed sample (W2261). (**a**) and (**b**) are BSE images. The red frame in (**a**) indicates where the region of (**b**) was picked. The majority of the grey phase is olivine, with small contrast caused by orientation. The oblate and rotund voids are $CO_2$ (formerly $CO_2$-filled). The voids at triple junctions are interpreted to be $CO_2$ and surrounding the carbonate melt, removed during polishing. A few CPx grains (light grey) are observed. The compositions of light-grey and dark-grey carbonate melt are provided in the Appendix A, Table 1. (**c**) Frequency distribution of grain size. The red curve is a lognormal distribution of grain size. The mean is noted on the histogram. (**d**) Pole figures of crystallographic orientations. Color scale indicates the multiples of a uniform distribution (MUD).

The grain size of the hot-pressed sample roughly fit to a lognormal distribution, with a mean grain size of 5.7 μm, as illustrated in Figure 2c. The CPO was approximately random with an M-index of 0.017, as illustrated in Figure 2d. For water content, the polycrystal of the hot-pressed sample (including grain boundaries) contained approximately 9 ppm $H_2O$ by weight (FTIR result presented in Appendix A). Chemical compositions of olivine,

CPx, and carbonate melt were also analyzed by electron microprobe (see Table 1 in the Appendix A for details and Appendix A).

### 3.2. Mechanical Data

The sample conditions and deformation conditions including the imposed strain rate, stress, and strain at the end of each experiment are summarized in Table 1. As described in the Methods section, samples were hot-pressed and then immediately deformed. After deformation, the sample was flattened normal to the imposed shear direction. Therefore, the strain rate of an experiment was calculated using the imposed displacement rate and interpolation of the sample thickness. No slip between pistons and samples was observed (Figure 1c).

**Table 1.** Summary of experiments.

| No. | Experiment Type | Measured Porosity | Axial Displacement Rate (mm/s) | Equivalent Strain Rate (/s) | Equivalent Stress (MPa) | Shear Strain | Equivalent Strain | Grain Size * (μm) | Thickness * (mm) |
|---|---|---|---|---|---|---|---|---|---|
| W2261 | Hot-pressing | 7.0% | | | | | | 5.7 | |
| W2263 | Rate stepping | 5.8% | $7.5 \times 10^{-5}$ | $6.0 \times 10^{-5}$ | 131 | 1.8 | 1.0 | | |
| | | | $1.9 \times 10^{-5}$ | $1.6 \times 10^{-5}$ | 61 | 3.0 | 1.7 | | |
| | | | $1.9 \times 10^{-4}$ | $1.9 \times 10^{-4}$ | 237 | 4.4 | 2.5 | | |
| | | | $7.5 \times 10^{-5}$ | $8.0 \times 10^{-5}$ | 147 | 4.9 | 2.8 | 4.5 | 0.74 |
| W2264 | Constant rate | 7.2% | $7.5 \times 10^{-5}$ | $5.9 \times 10^{-5}$ | 178 | 3.2 | 1.8 | 5.8 | 0.88 |
| W2266 | Annealing | 6.5% | $7.5 \times 10^{-5}$ | $6.7 \times 10^{-5}$ | 174 | 3.2 | 1.8 | 7.3 | 0.77 |

*: Grain size and thickness were measured after the experiment.

Graphs of equivalent stress plotted against equivalent strain are presented in Figure 3. All of the curves for the experiments showed a rapid stress rise to a peak stress at an approximate equivalent strain of ~0.2, followed by a steep drop to a more slowly changing stress with increasing strain. The experiments shown in Figure 3a were carried out at the same equivalent strain rate of ~$6.0 \times 10^{-5}$ s$^{-1}$ and the equivalent stresses were around 175 MPa. The shear strain calculated using the displacement measure with the external linear variable displacement transducer (LVDT) ($\gamma = 3.2$) is consistent with the offset of the piston (Figure 1c) after correction for the contribution of the elastic loading of column. Four strain-rate steps were performed during the rate-stepping experiment. During each strain-rate step, when the stress reached a steady state, the strain rate was changed to the next step. Although strain-rate steps were taken in an experiment, a flow law was not derived based on only four data points, because the number of parameters in a flow law will render the problem underdetermined; a discussion of a flow law for our aggregates consisting of olivine, CPx, $CO_2$ and carbonate melt is beyond the scope of this paper. Nonetheless, we note that the apparent stress exponent, $n = 1.8$, based on data from experiment W2263, is similar to that reported by Qi et al [10], suggesting a significant component of dislocation creep.

### 3.3. Distribution of $CO_2$

Here, detailed microstructural observations of a deformed sample (W2264) are presented. As illustrated in Figure 4, $CO_2$ and carbonate melt were highlighted as pores in the micrographs of the etched polished section. The size and distribution of the pores illustrates the redistribution of $CO_2$ induced by shear deformation. The size of the pores, that is, the size of $CO_2$ and melt pockets, ranged from a couple of microns to close to the lower limit of the olivine grain size. The smaller pores, probably single $CO_2$ pockets, were sparsely and homogenously distributed at olivine three- and four-grain junctions and along the olivine grain boundaries as isolated "bubbles". The presence of $CO_2$ pocket at triple junctions can arise because of a relatively high fraction of $CO_2$ (>5%), which is consistent with the calculations of von Bargen and Waff [42].

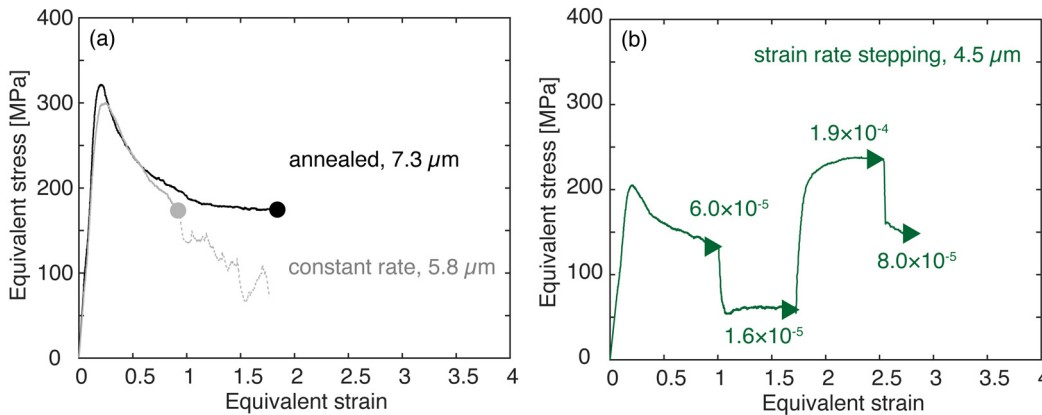

**Figure 3.** Mechanical data for the deformation experiments on $CO_2$-bearing samples with final grain size and experiment type noted for each curve. (**a**) Equivalent stress versus equivalent strain plots for general shear experiments conducted at the strain rate of ~$6.0 \times 10^{-5}$ s$^{-1}$; circles illustrated where mechanical data were determined. The grey and black lines illustrate results for the experiments (W2264 and W2266, respectively). Sample W2266 was annealed for 10 h at 1 GPa and 1100 °C after deformation. At the end of the experiment W2264, the deformation piston cracked and influenced the stress data (dashed lines), and thus we report mechanical data (grey circle) from before this happened. (**b**) Equivalent stress versus equivalent strain plot for rate-stepping experiment (W2263) deformed in general shear. Strain rates (s$^{-1}$) are noted for each step.

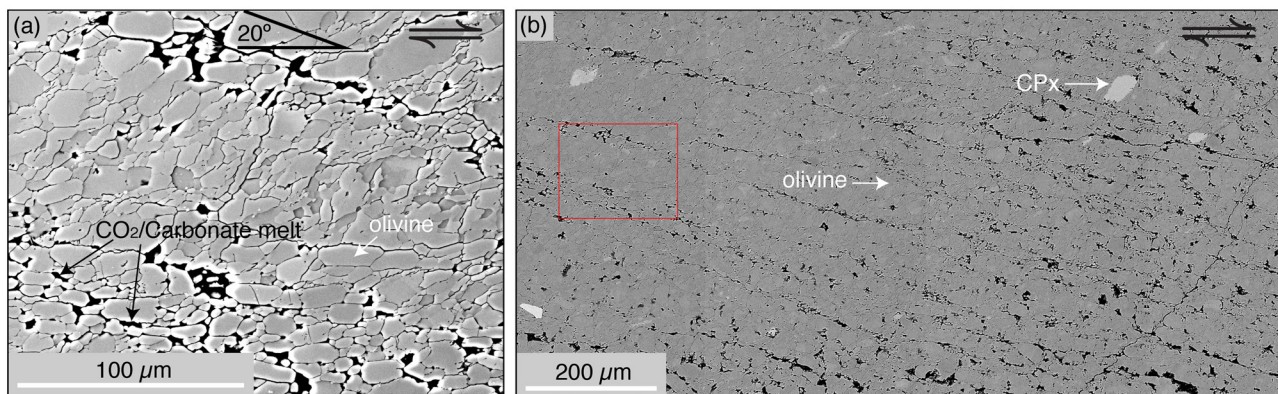

**Figure 4.** BSE micrographs of the etched surface of a deformed sample, W2264. (**a**) A close look at $CO_2$-enriched bands and the $CO_2$-depleted region in between. The darkest regions with bright rims are $CO_2$ pockets/layers, but they may appear to be larger due to etching. (**b**) A broad view of the deformed sample. The red frame indicates where the region of (**a**) was picked. The diopside (light phase) is set in the olivine grains (gray). The tubules (black) are formerly $CO_2$-filled.

The distribution of $CO_2$ and melt was dramatically changed after deformation. $CO_2$ segregated into high-porosity bands, that is, $CO_2$-enriched bands, orienting ~20° from the shear plane, synthetic to the imposed shear direction. $CO_2$-enriched bands, which occurred across the entire section, were separated by low-porosity regions, that is, $CO_2$-depleted regions. Within the bands, pores were interconnected. The $CO_2$-enriched bands were ~30 μm wide, while the $CO_2$-depleted regions were ~100 μm wide.

### 3.4. Crystallographic Preferred Orientation and Shape Preferred Orientation

To examine the crystallographic preferred orientation and shape preferred orientation of olivine grains in our $CO_2$-bearing samples, pole figures highlighting the CPOs and rose diagrams quantifying the SPOs are presented in Figure 5. In both deformed samples, CPOs are characterized by girdles of [100] and [001] axes sub-parallel to the shear direction, and strong point maxima of [010] axes sub-perpendicular to the shear plane, namely the AG-type fabric [11]. The fabric is rotated by 20 to 30°, antithetic to the shear direction, from

[010] axes being normal to the shear plane. With equivalent strain increasing from 1.8 to 2.8, the strength of the fabric increased slightly, with the J-index increasing from 2.39 to 2.77 and the M-index increasing from 0.135 to 0.158. In both deformed samples, the long axes of olivine grains align $27 \pm 3°$ from the shear plane, antithetic to the shear direction.

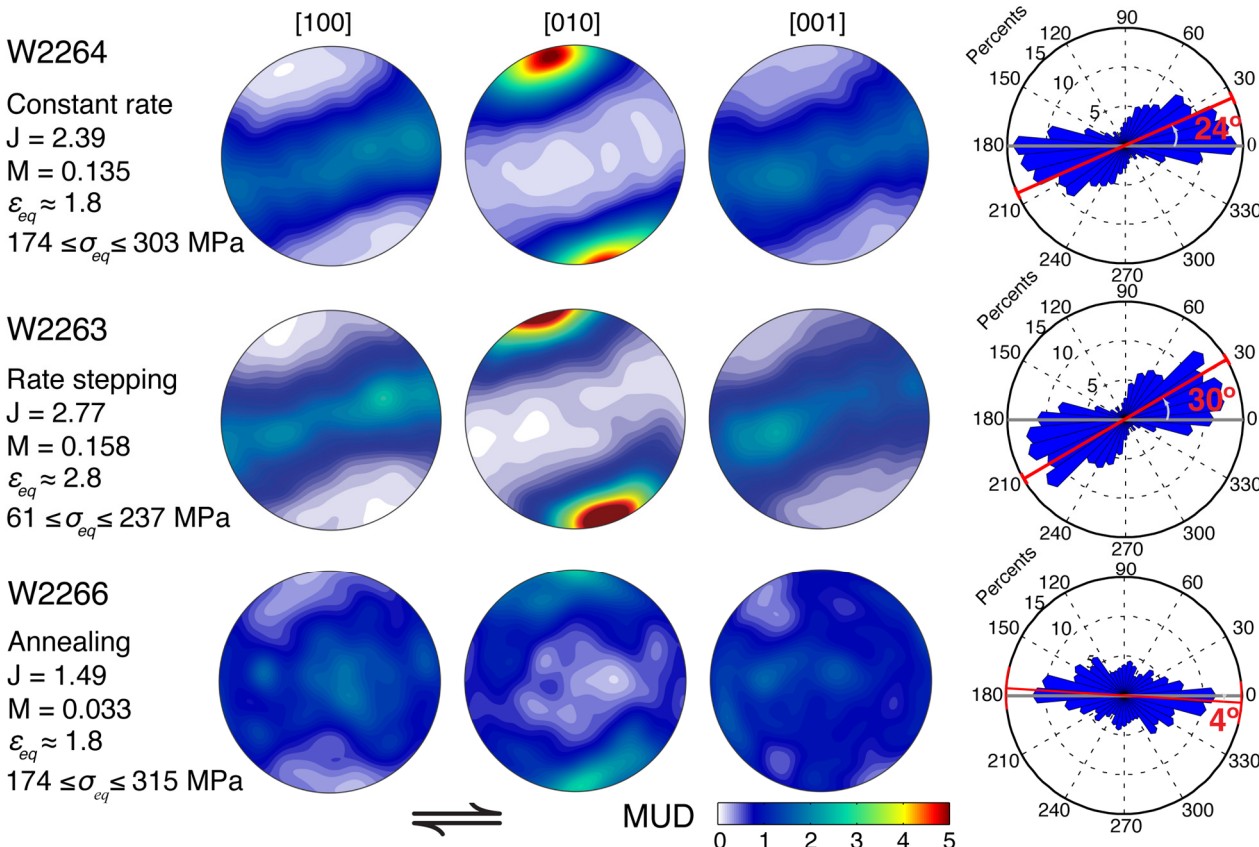

**Figure 5.** CPOs and SPOs in the profile plane of 3 samples. From top to bottom are the sample deformed at a constant rate, W2264, the sample deformed at different strain rate steps, W2263, and the sample annealed for 10 h after deformation W2266. Pole figures and rose diagrams were processed from grains with grain areas greater than 2 $\mu$m$^2$ and aspect ratios greater than 1.2. The J-index and M-index were calculated for each crystallographic fabric. The angle between the SPO and the shear direction is noted on each rose diagram.

The annealed sample was very different from the two deformed samples in terms of CPO and SPO. The CPO was characterized by a diffused cluster of [010] axes normal to the shear plane, and almost random distributed [100] and [001] axes. This CPO was very weak, with a J-index of 1.49 and an M-index of 0.033. The long axes of olivine grains aligned roughly in the shear plane.

### 3.5. CPOs in and out of a CO$_2$-Enriched Band

To examine the influence of the CO$_2$-enriched bands on CPO formation, pole figures were processed from a CO$_2$-enriched band region and a CO$_2$-depleted non-band region in the profile plane deformed sample. The comparison shown in Figure 6 of the pole figures from these two regions reveals that the general orientations and distributions of crystallographic axes are little different in the two regions. In terms of the J-indexes and M-indexes, the strength of the crystallographic fabric in the band region is somewhat stronger than that in the entire region in Figure 6, and accordingly, the non-band region shows a weaker CPO than the entire region. In addition, the girdles of [100] and [001] axes, the strong [010] (maximum normal to the girdle), and the rotation of the fabric are similar in the entire region and non-band region. The 30° back-rotation is observed in both

entire region and non-band regions. By contrast, the back-rotation in the band regions is ~0°, indicating that the deformation kinematics are different between non-band and band regions.

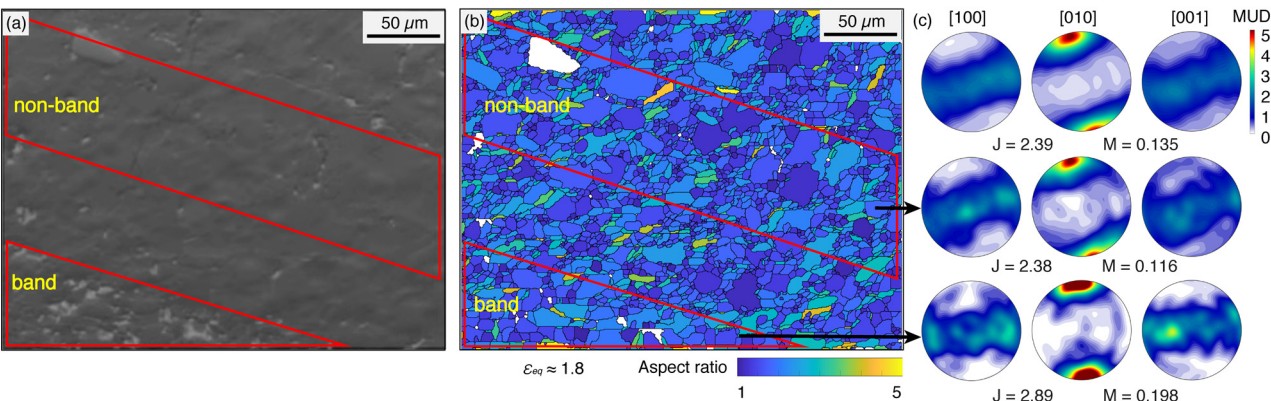

**Figure 6.** Comparison of CPOs in a $CO_2$-enriched "band" region and in a $CO_2$-depleted "non-band" region in the profile section of deformed sample (W2264). (**a**) Electron image. (**b**) The orientation map is colored by the aspect ratios of grains. White pixels are unindexed points. The parallelogram-shaped region noted as "band" is a $CO_2$-enriched band. The parallelogram-shaped region noted as "non-band" is a $CO_2$-depleted region. (**c**) From top to bottom, the three sets of pole figures are for the entire region, non-band region, and band region, respectively. Pole figures are universally colored by their multiples of uniform distributions (MUD), which is illustrated in the color bar. The maximum and minimum values of MUDs are listed for each pole figure. The sense of shear for the pole figures is top to the right. J- and M-indexes are included for each region.

## 4. Discussion

### 4.1. The Formation of $CO_2$-Enriched Bands

Segregation of the fluid phase occurred during deformation of our samples, as illustrated in Figure 4. Fluid/melt-enriched bands formed in a similar manner to those reported by Holtzman [2] and King et al. [5]. In contrast to previous studies, in which basaltic melt was the fluid phase, in our samples, the fluid phase consisted of carbonate melt and supercritical $CO_2$. The dihedral angle of $CO_2$ was ~80–90°, too large to form an interconnected fluid network. However, the dihedral angle of carbonate melt was much smaller, at ~30°, which is sufficient for the formation of an interconnected fluid network and thus the migration of the fluid phase. However, if only carbonate melt could migrate, $CO_2$ should maintain a roughly uniform distribution. As illustrated in Figure 4, not many $CO_2$ pores existed in the fluid-depleted regions, suggesting $CO_2$ may migrate together with carbonate melt. Further investigations into such two-phase systems are necessary to examine this interpretation [43].

### 4.2. A Comparison with the CPOs in Olivine + Basalt

Our observations on sheared samples revealed an AG-type fabric in $CO_2$-bearing olivine aggregates. Compared with the A-type fabric, typically observed in anhydrous olivine aggregates sheared at similar conditions to this study [44], the AG-type fabric is characterized by two key features: (1) girdles of [100] and [001] axes at a low angle to the shear plane, with a strong [010] maximum normal to the girdle, and (2) antithetic rotation of the fabric with respect to the shear plane. The CPO is very similar to the CPOs observed in sheared partially molten rocks of olivine and basalt [4,10], except for the fact that the weakly aligned, shear-direction-normal [100] axes in the shear plane were not observed in our samples. Qi et al. [10] proposed a melt-assisted, SPO-induced CPO model to explain the formation of the CPOs in the olivine + basalt samples. Based on the similarities and differences in the CPOs between sheared $CO_2$-bearing olivine and basaltic melt-bearing olivine, in the following discussions, we examine the applicability of the model to our samples and explain the formation of the CPOs.

### 4.3. Crystal Habit of Olivine Grains

One key in the SPO-induced CPO model is that the grain shape is crystallographically controlled [45]. Studies of the morphology of olivine grains in an ultramafic melt demonstrate a crystal habit with grain shapes of which the longest are parallel to the [001] axis and the shortest are parallel to the [010] direction [46], which was observed in the partially molten samples analyzed by Qi et al. [10]. However, the crystal habit of olivine grains in the $CO_2$-bearing samples, that is, a solid-fluid system with supercritical $CO_2$ and carbonate melt as the fluid phase, is unknown.

To explore the morphology of olivine grains in the $CO_2$-bearing samples, grain shape and crystallographic orientation were examined for two orthogonal surfaces (profile and shear planes) on a $CO_2$-bearing sample sheared to a strain of 1.8. In Figure 7, the orientations of crystallographic axes of grains selected based on their grain shape in the section are examined with pole figures, rose diagrams of the orientations of long axes of grains, and histograms of aspect ratios. In the pole figures, light gray girdles and clusters in the background represent the overall CPO observed in this section. The yellow stars indicate the reference orientation for the subset of grains used to make the other plots in this panel. Blue and red dots in the pole figures indicate the orientations of the selected subset of grains. The orientations of these grains lie within 30° of the target orientation. Each grain in a subset is fitted with an ellipse with its long-axis orientation plotted in the rose diagram and aspect ratio plotted in the histogram. This information permits a comparison of the preferred orientations of crystallographic axes with the orientation of long axes of grains.

In the profile plane, [010] axes form strong point maxima sub-perpendicular to the shear plane, so comparisons were only obtained between the relative lengths of grains along the [001] and [010] axes (Figure 7a), and between the relative lengths of grains along the [100] and [010] axes (Figure 7b). In Figure 7a, [001] axes align subparallel to the shear direction in this subset, and the SPO reveals that the orientation of the long axes is in the same direction as the [001] axes. Therefore, grains are longer along [001] axes than along [010] axes. Similarly, Figure 7b indicates that grains are longer along [100] axes than along [010] axes. In the shear plane (Figure 7c,d) the relative lengths along the [100] and [001] axes are compared. In both panels, grain shape exhibits a bimodal distribution. The primary mode corresponds to long [001] axes, while the secondary mode corresponds to elongated [100] axes. This result suggests that grains have similar lengths along [100] and [001] axes, with more grains slightly longer parallel to [001] axes.

Note that this crystal habit is different from that reported in Miyazaki et al. [45], in which forsterite exhibited elongation along the [100] axis, or Qi et al. [10], in which San Carlos olivine exhibited elongation along the [001] axis. These differences suggest that the olivine crystal habit can vary with the composition of the fluid phase and the deformation conditions.

### 4.4. SPO-Induced CPO Model

CPO can be induced by a crystallographically controlled SPO [45]. As elaborated in Qi et al. [10], during simple shear, the long axes of elongated grains are rotated towards the shear direction, which produces an SPO. Then, if the elongated grain shape is crystallographically controlled, the SPO results in a CPO. As demonstrated in the previous subsection, olivine grains are on average shortest along [010] axes and similar in length along [100] and [001] axes. Simple shear of these oblate ellipsoidal grains would align the shortest axes, the [010] axes, normal to the local shear plane, resulting in a cluster of [010] axes, and homogeneously distribute the two similarly long axes, the [100] and [001] axes, in the local shear plane, resulting in girdles of [100] and [001] axes. The 30° back-rotation observed in both CPO and SPO is consistent with the SPO-induced CPO model.

### 4.5. Local Deformation Geometry

Fluid phase segregation occurred in our sheared $CO_2$-bearing samples, in a very similar way to the melt segregation that occurred in sheared olivine + basalt samples. When

stress-driven melt segregation occurred, an anastomosing network of melt-enriched bands formed at an angle to the bulk shear plane, synthetic to the bulk shear direction. These melt-enriched low-viscosity bands deformed more easily than melt-depleted regions. Because of strain incompatibility at the sample ends, the imposed shear requires the non-band regions to accommodate strain at an angle to the bulk shear plane, rotated antithetically to the imposed shear direction [47]. Similarly, this strain partitioning process must apply to sheared $CO_2$-bearing samples, where stress-driven phase segregation occurred. Therefore, the counterclockwise rotation of [010] axes from the normal to the shear plane is due to strain partitioning between the $CO_2$-enriched and $CO_2$-depleted regions. The [010] axes are normal to the local shear plane, and [100] and [001] axes lie in the local shear plane in the $CO_2$-depleted portions of a sample, as the bands only take a small fraction of the total volume.

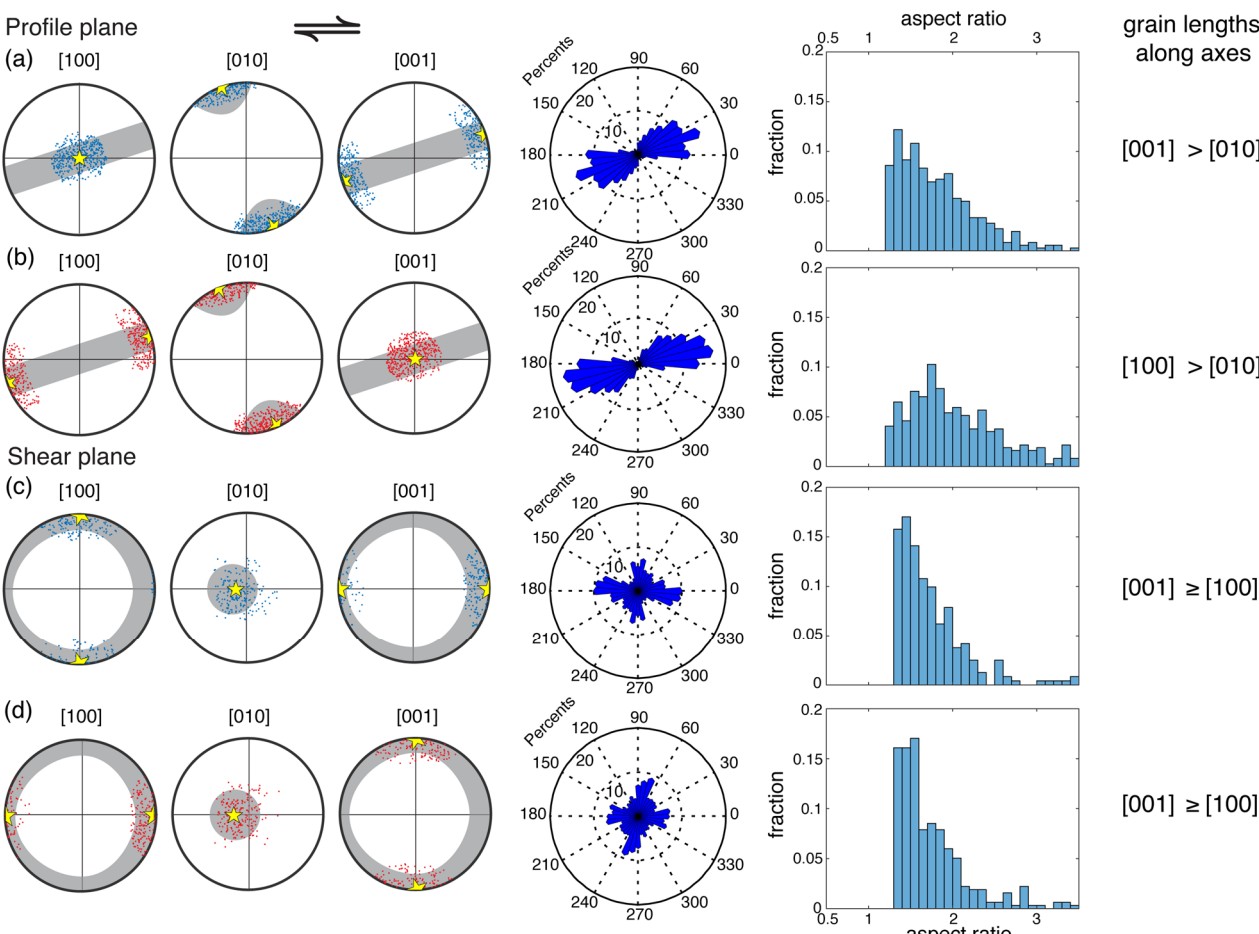

**Figure 7.** For the deformed sample, W2264, comparison of orientation of crystallographic axes with the direction of the long axes of selected subsets of grains. In plots (**a–d**), the blue and red points in the pole figures indicate the subsets of grains that were selected based on their crystallographic orientations, the rose diagrams show the orientations of the long axes of this subset of grains, and the histograms illustrate their aspect ratios. The gray regions in the pole figures represent the overall CPO, and the yellow stars represent the reference orientation of the subset of grains. In the rose diagrams, the length of each bar corresponds to the fraction of grains with a specific orientation. The relative lengths of grains along the three crystallographic axes are summarized in the far right-hand column. Only grains with aspect ratios greater than 1.2 were used in this analysis.

### 4.6. CPO Modification during Annealing

Grain size and the CPO of the annealed sample were compared with those of the deformed samples in Table 1 and Figure 5. After annealing, the grain size increased from

5.8 μm to 7.3 μm (increased by 26%) in 10 h. Grain growth in this sample was much slower than that in the samples of Boneh et al. [48], in whose study 200 μm grains were observed after only four hours of annealing. We infer that the distribution of $CO_2$ pockets in the deformed samples (Figure 4) probably inhibits the rate of grain boundary migration, thus limiting the grain size in the annealed sample. $CO_2$-enriched bands were less obvious in the annealed sample (Appendix A).

The CPO of the annealed $CO_2$-bearing samples was modified significantly in terms of its strength and its orientation. The M-index decreased from 0.135 to 0.033 after 10 h of annealing, and the decrease rate was significantly faster than that reported by Boneh et al. [48]. Compared to the deformed samples, the cluster of the [010] axes rotated to become perpendicular to the shear plane. Shape preferred orientation also changed after annealing, with the long axes of olivine grains aligning roughly parallel to the shear direction (Figure 5). The rotation of the CPO correlated with the rotation of the SPO. These data are insufficient to explain the changes in the microstructure during annealing, which were dramatically different from the changes in melt-free samples, but the alteration of the interfacial energy due to the existence of fluid/melt phase may significantly affect the annealing behavior.

## 5. Conclusions

We investigated the microstructure in sheared $CO_2$-bearing olivine aggregates. Stress-driven fluid segregation occurred, producing $CO_2$-enriched bands at ~20° to the shear plane, synthetic to the shear direction. An AG-type fabric was found in sheared samples. [100] and [001] axes formed girdles sub-parallel to the shear plane, and [010] axes formed a strong cluster sub-normal to the shear plane. A strong correlation existed between the orientation of the gridles and the SPO, indicating that this CPO was developed due to the SPO-induced CPO model. Our study suggests that the CPO formation mechanism established in the solid-fluid system of olivine + basalt also applies to the solid-fluid system of olivine + $CO_2$. Degassing of $CO_2$ from peridotite is an important part of global $CO_2$ cycling. Our results provide constraints on the microstructures effect of $CO_2$ on peridotite and the relative of CPO and SPO.

**Author Contributions:** Conceptualization, X.H. and G.H.; methodology, G.H., N.Z. and H.Z.; validation, H.Z., N.Z., C.Q., X.H. and G.H.; formal analysis and investigation, H.Z., N.Z., G.H. and X.H; resources, H.Z.; data curation, H.Z. and N.Z; writing—original draft preparation, H.Z.; writing—review and editing, N.Z., G.H. and C.Q.; visualization, H.Z., N.Z. and C.Q.; supervision, G.H., C.Q. and X.H.; project administration, G.H. and X.H.; funding acquisition, G.H. and X.H. All authors have read and agreed to the published version of the manuscript.

**Funding:** This research was funded by NSF grant EAR-1624178 and NSFC grants 41574089 and 41774096.

**Data Availability Statement:** The data for this study are available from the Brown Digital Repository (https://doi.org/10.26300/g0wk-ty19, accessed on 28 December 2020).

**Acknowledgments:** This work benefited greatly from discussions with Reid Cooper, Terry Tullis, Yuval Boneh, Eric Burdette, Nir Badt, Leif Tokle, and Pamela Speciala at Brown University. We appreciate Seth Krukenburg for providing access to EBSD at Boston College, and Kevin Robertson for providing access to FTIR at Brown University. We thank Yongsheng Zhou, Xi Ma and Jiaxiang Dang for providing help with EBSD at the Institute of geology China Earthquake Administration.

**Conflicts of Interest:** The authors declare no conflict of interest.

## Appendix A

Contents of this file: Appendix A Figures A1–A6 and Table 1.

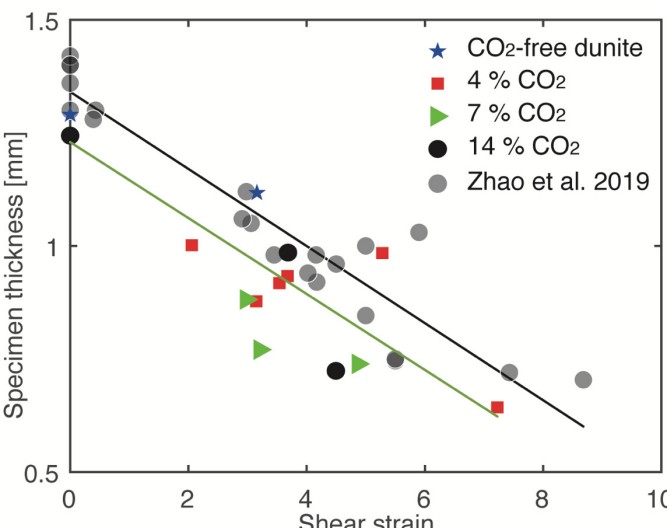

**Figure A1.** The thickness of the specimens as a function of the shear strain. The result includes data from different $CO_2$ contents (4%, 7%, 14%) and $CO_2$-free dunite. Gray circles are data from Zhao et al. [27]. The green line is a least square fit to data from this work; the black line is a least square fit to the data from Zhao et al. [27].

Water contents in the polycrystalline aggregates were determined by Fourier transform infrared (FTIR) spectroscopy in transmission. To calculate hydroxyl concentration, we integrated spectra between 3650 $cm^{-1}$ and 3000 $cm^{-1}$, which covered infrared hydroxyl stretching bands for Olivine, CPx, and OPx [27,49–51].

We examine the hot-pressed sample to obtained water contents; the thickness of the sample wafer was 0.3 mm. The polycrystal of the hot-pressed sample (including grain boundaries) contained ~9 ppm $H_2O$ by weight (~150 $H/10^6$ Si), indicating that olivine aggregates are nominally anhydrous prepared through this procedure. FTIR results are presented in Figure A2.

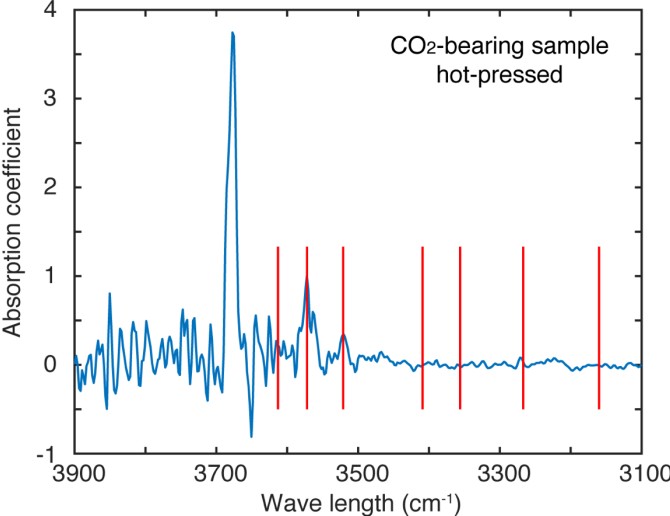

**Figure A2.** FTIR spectra from hot-pressed polycrystalline sample (W2261). The characteristic OH stretching bands in Ol (3613, 3572, 3521, 3409, 3356, 3267, 3160 $cm^{-1}$) were presented by solid red lines.

EPMA shows the composition of the starting sample and deformed sample. The results of Table 1 show the composition of the dark gray melt in Figure 2a,b and the light gray melt at Figure 2b, and the light gray melt between sample and piston (Figure A3).

Images of Figure A3 obtained from an un-etched deformed sample. The light phase at triple junctions is Ca-rich carbonate melt and the dark phase is Mg-rich carbonate melt. Figure A3a,b show reaction of the sample with the alumina piston.

**Table 1.** Summary of the EPMA.

| No. | CaO | MgO | FeO | SiO$_2$ | Al$_2$O$_3$ | Cr$_2$O$_3$ | TiO$_2$ | Na$_2$O | K$_2$O | MnO | NiO | CO$_2$ | Total | Comment |
|---|---|---|---|---|---|---|---|---|---|---|---|---|---|---|
| W2261-light [1]-gray-melt | 48.17 | 7.21 | 1.54 | 5.84 | 0.00 | 0.00 | 0.00 | 0.01 | 0.01 | 0.03 | 0.05 | 37.17 | 100.00 | Hot-pressed specimen |
| W2261-dark [2]-gray-melt | 1.17 | 36.89 | 5.88 | 5.66 | 0.03 | 0.00 | 0.03 | 0.05 | 0.00 | 0.08 | 0.14 | 50.08 | 100.00 | Hot-pressed specimen |
| W2264-dark [3]-gray-melt | 1.42 | 45.45 | 7.75 | 38.42 | 0.60 | 0.20 | 0.02 | 0.03 | 0.00 | 0.09 | 0.37 | 5.64 | 99.99 | Deformed specimen |
| W2264-light [4]-gray-melt | 23.98 | 19.99 | 2.13 | 48.93 | 6.30 | 0.02 | 0.09 | 0.10 | 0.00 | 0.04 | 0.13 | 0.00 | 101.75 | Deformed specimen |
| Dolomite | 34.40 | 23.08 | 0.02 | −0.09 | 0.00 | 0.01 | −0.01 | 0.00 | 0.01 | 0.00 | −0.02 | 42.28 | 99.88 | Dolomite |
| W2261-olivine | 0.13 | 49.24 | 8.60 | 40.04 | 0.01 | 0.02 | 0.02 | −0.01 | −0.01 | 0.13 | 0.43 | 1.34 | 99.96 | Hot-pressed specimen |
| W2261-diopside | 13.34 | 30.36 | 5.12 | 47.19 | 3.14 | 0.40 | 0.37 | 0.27 | 0.00 | 0.12 | 0.19 | 0.00 | 100.49 | Hot-pressed specimen |

Light [1]: is the light triple junctions of Figure 2b; Dark [2]: is the dark gray melt of Figure 2a,b; Dark [3]: is the dark gray melt of deformed sample not etched; Light [4]: is the light gray melt between sample and piston of Figure A3c.

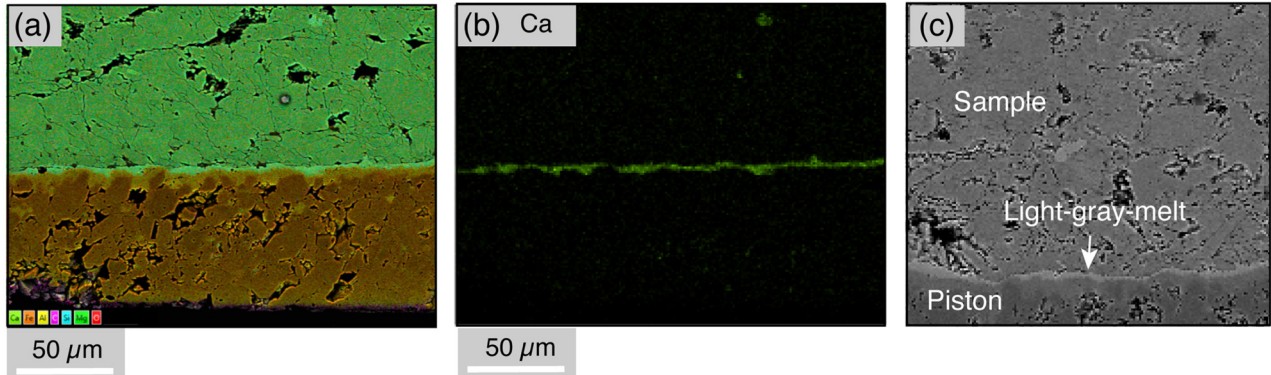

**Figure A3.** The sample section for the deformed sample (W2264, not etched). (**a**,**b**) The energy-dispersive X-ray spectroscopy (EDS). (**c**) A SEM image. The composition of light-gray-reaction phase between the sample and the piston was analyzed by EPMA.

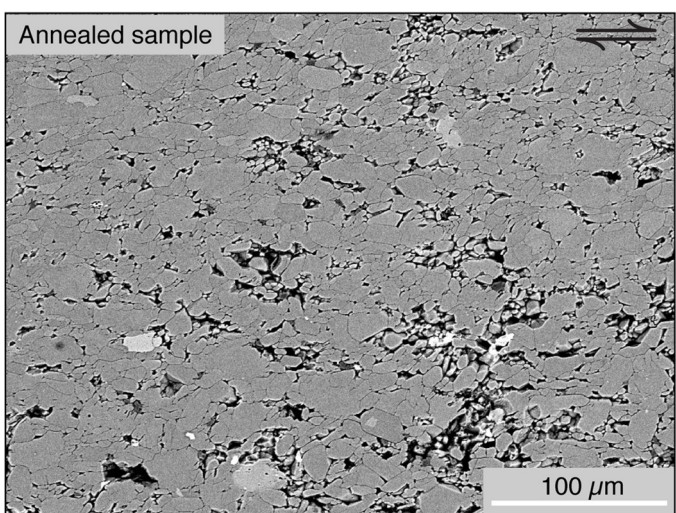

**Figure A4.** The BSE image of the annealed sample (W2266) after etching.

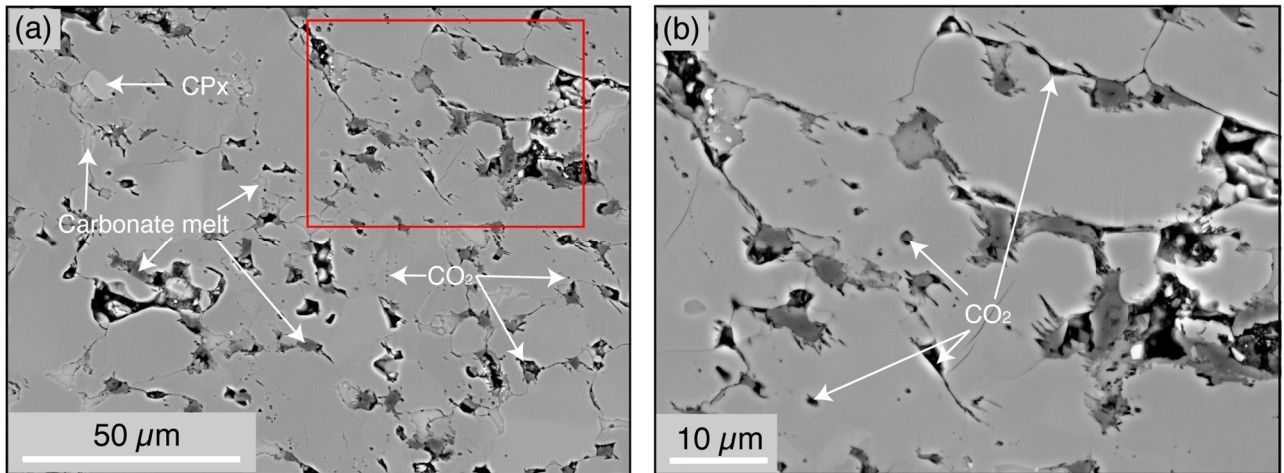

**Figure A5.** (**a**) The BSE images of the annealed sample (W2261, Figure 2). The majority of the grey phase is olivine. (**b**) The oblate and rotund voids are $CO_2$ (formerly $CO_2$-filled) with higher magnification. The voids at three- and four-grain junctions are interpreted to be $CO_2$. The red frame in (**a**) indicates where the region of (**b**) is picked.

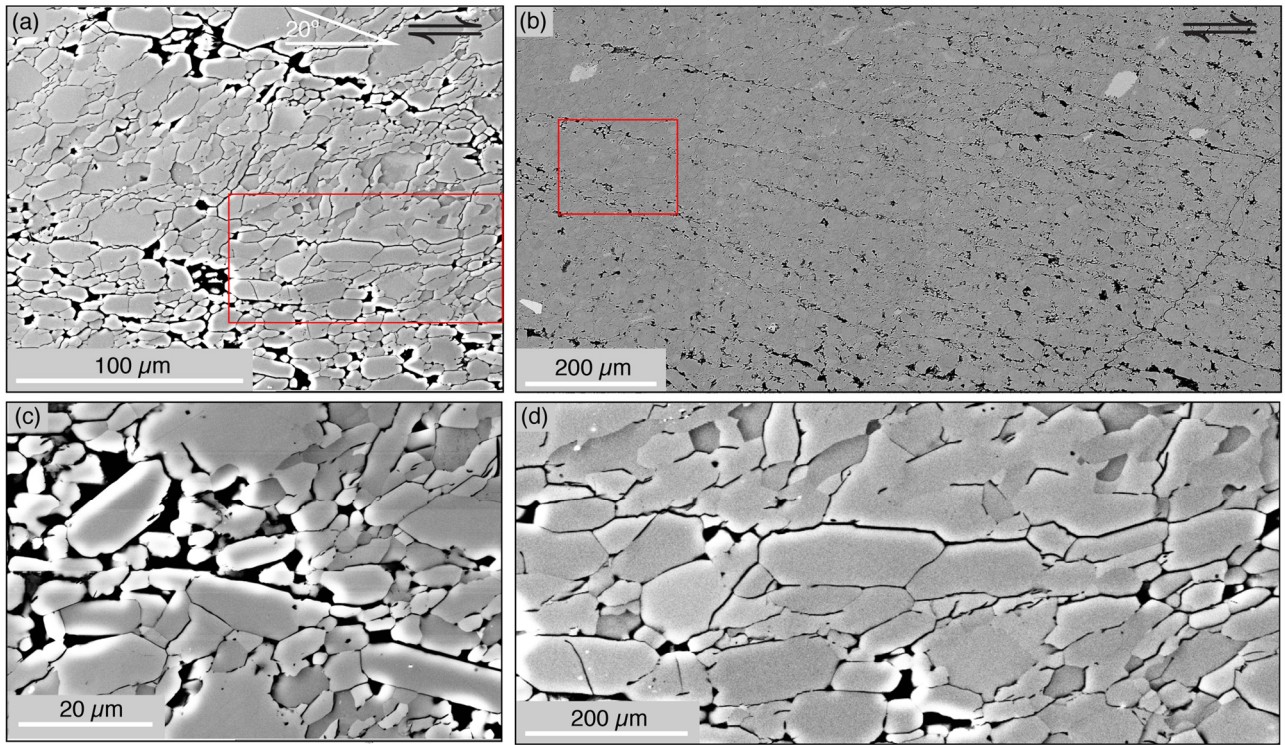

**Figure A6.** (**a**,**b**) The BSE images of the Figure 4. (**c**,**d**) The close look at $CO_2$-enriched bands and the $CO_2$-depleted region. The red frame in (**a**) indicates where the region of (**d**) is picked. The red frame in (**b**) indicates where the region of (**a**) is picked.

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
