# Peer review of "Influences of CO2 on the Microstructure in Sheared Olivine Aggregates"

_minerals, doi:10.3390/min11050493_

Round 1
Reviewer 1 Report
Review on the manuscript MINERALS-1192191 “Influences of CO2 on the microstructure in sheared olivine”, by Zhang et al.
General comments
This manuscript investigates the effects of shear deformation process of a solid-fluid system on microstructure development in the mantle analogues. In their experimental runs at 1GPa and 1100°C, olivine + dolomite mixture is subjected to shear using a Griggs-type apparatus. After high temperature and high pressure shearing, dolomite decomposes into carbonate melt and CO2 and an AG-type fabric is observed. Such results have similarities with olivine + basalt experiments from the literature, so the authors conclude that similarly to a basaltic melt, CO2 influences the microstructure development and can be explained by the crystallographic preferred orientation (CPO) formation model for the solid-fluid system.
I believe that this a good contribution to an important topic however some improvements are necessary to clarify the manuscript. I recommend the acceptance of the manuscript after revisions. See my comments below.
Specific comments
Line 18 – replace “as” by “if”.
Line 21 – replace “…fabric occurred...” by “…fabric is observed...”
Line 22 – I don’t understand why you use the word “inconsistency” here, because in the line below you conclude that the microstructures are similar to a that of a basaltic melt. Please rephrase (consistent?).
Line 44-67 – This paragraph is too long please reword and divide it appropriately.
Line 60 – “…CO2 and carbonate melt are also important for mantle dynamics.”
Line 77 – Rephrase “..powders initially containing olivine…”
Line 181 – The presence of “blisters” shows that CO2 migrated/segregated from the experimental matrix. I wonder if after segregation the pores on the grain boundaries could somehow shrink or close. Can you explain if you see any evidence for that?
Lines 190-192 – I think it would be useful to see close-up images of the triple junctions with carbonate melt and the ones with the voids (interpreted as former sites with CO2). Also in Figure 2 CO2 is also pointed as small cavities in inside the olivine grains. Could they be fluid inclusions? Again I think that images with higher magnification would be good here.
Line 192-193 – Experiments on partially melted granites (Rosenberg and Handy 2001 – journal of metamorphic geology) show that an interconnected network of melt is formed at a 7% melt threshold on the grain boundaries, allowing the melt to be extracted from the framework. I know here you are dealing with different materials, but I wonder if the 7% porosity in your experiments may also be meaningful. Can you add something on that? Maybe there is some literature on it (how much of fluid would be need to form the network). My point is that, as you described in the introduction, wetting angles of CO2 are too large (>60°) and as result an interconnected network is unlikely to form (Watson and Brenan 1987 – EPSL). However, your experiments show that with the deformation CO2-enriched bands can be developed and by looking that Figure 4a we can see that the bands are interconnecting. Please develop these thoughts on the new version.
Figure 4 – Please label the minerals, carbonate melt and inferred CO2. And add some high magnification images to show more details on the interconnected pores. I see that some pores away from the Co2-enriched bands do have high dihedral angles, and at the same time within the CO2-enriched bands it’s possible to observe the pores coalescing to form larger pockets. Those things are interesting for the reader!
Line 258 – Replace “In” by “When”.
Line 258 – Rephase “..melt mostly tended to aggregate..”
Author Response
The comments are noted in black, and the replies are noted in blue.
Reviewer # 1:
General comments:
This manuscript investigates the effects of shear deformation process of a solid-fluid system on microstructure development in the mantle analogues. In their experimental runs at 1GPa and 1100°C, olivine + dolomite mixture is subjected to shear using a Griggs-type apparatus. After high temperature and high pressure shearing, dolomite decomposes into carbonate melt and CO2 and an AG-type fabric is observed. Such results have similarities with olivine + basalt experiments from the literature, so the authors conclude that similarly to a basaltic melt, CO2 influences the microstructure development and can be explained by the crystallographic preferred orientation (CPO) formation model for the solid-fluid system.
I believe that this a good contribution to an important topic however some improvements are necessary to clarify the manuscript. I recommend the acceptance of the manuscript after revisions. See my comments below.
Specific Comments
Line 18 – replace “as” by “if”.
Re: revised as suggested.
Line 21 – replace “…fabric occurred...” by “…fabric is observed...”.
Re: revised as suggested.
Line 22 – I don’t understand why you use the word “inconsistency” here, because in the line below you conclude that the microstructures are similar to a that of a basaltic melt. Please rephrase (consistent?).
Re: We changed “inconsistency” to “similar to”.
Line 44-67 – This paragraph is too long please reword and divide it appropriately.
Re: We divided the “Introduction paragraph” into 2 paragraphs.
Line 60 – “…CO2 and carbonate melt are also important for mantle dynamics.”
Re: revised as suggested.
Line 77 – Rephrase “…powders initially containing olivine…”
Rephrase: Sample powders contain 88 wt% olivine, 3 wt% clinopyroxene (CPx, with ~8 wt% of orthopyroxene; [26]) and 9 wt% dolomite.
Line 181 – The presence of “blisters” shows that CO2 migrated/segregated from the experimental matrix. I wonder if after segregation the pores on the grain boundaries could somehow shrink or close. Can you explain if you see any evidence for that?
Re: We did not observe evidence for shrinking of pores on the grain boundaries (e.g., no increase of grain orientation spread around pores). This is probably because, in addition to the volume increase resulting from the return to ambient conditions, the olivine matrix is too strong to deform (either brittly or plastically) due to the release of gas pressure at room temperature.
Lines 190-192 – I think it would be useful to see close-up images of the triple junctions with carbonate melt and the ones with the voids (interpreted as former sites with CO2). Also, in Figure 2 CO2 is also pointed as small cavities in inside the olivine grains. Could they be fluid inclusions? Again, I think that images with higher magnification would be good here.
Re: We added a high-magnification micrograph showing CO2 pockets in the appendix A5. The black dots shown in Figure 2 are actually along the grain boundary. The CO2 pockets may appear to be “inside” grains because grain boundaries are not well resolved in the unetched sample shown in Figure 2b. Likewise, in the etched sample we show the CO2 bubbles along grain boundaries (Figure 4). Also, our FTIR results indicate that the sample is far from water saturation (< 10 ppm), so water-rich fluid is unlikely to exist as a separate phase.
Line 192-193 – Experiments on partially melted granites (Rosenberg and Handy 2001 – journal of metamorphic geology) show that an interconnected network of melt is formed at a 7% melt threshold on the grain boundaries, allowing the melt to be extracted from the framework. I know here you are dealing with different materials, but I wonder if the 7% porosity in your experiments may also be meaningful. Can you add something on that? Maybe there is some literature on it (how much of fluid would be need to form the network). My point is that, as you described in the introduction, wetting angles of CO2 are too large (>60°) and as result an interconnected network is unlikely to form (Watson and Brenan 1987 – EPSL). However, your experiments show that with the deformation CO2-enriched bands can be developed and by looking that Figure 4a we can see that the bands are interconnecting. Please develop these thoughts on the new version.
Re: The porosity of 7 % was not chosen with any specific purpose. Indeed, different melts may form an interconnected network at different melt fractions owing to wetting characteristics. The reviewer is correct that the wetting angle of CO2is too large to form an interconnected network at low fluid fraction, but the connection is possible at fluid fractions ~5% (Waff & Bulau, 1979). In addition, with the presence of carbonate melt, which has a small wetting angle, melt could be interconnected at some scale. We added a new subsection (4.1) to explain this in the revised manuscript in Lines 334-346. We added some clarification about the presence of the CO2 and the melt phase, noting that individual pockets appear to be either melt or CO2; this suggests that neither the melt nor the CO2 are interconnected at the sample scale. Further work is needed to explore these ideas.
References: Waff, H.S.; Bulau, J.R. Equilibrium fluid distribution in an ultramafic partial melt under hydrostatic stress conditions. 1979, 84.
Figure 4 – Please label the minerals, carbonate melt and inferred CO2. And add some high magnification images to show more details on the interconnected pores. I see that some pores away from the CO2-enriched bands do have high dihedral angles, and at the same time within the CO2-enriched bands it’s possible to observe the pores coalescing to form larger pockets. Those things are interesting for the reader!
Re: We labeled the minerals and described where CO2 is in the caption for Figure 4 (Lines 281-289). Carbonate melt is not observable in Figure 4 due to acid etching. We noted that the CO2 is segregated due to deformation in the original version (Lines 275-280). We added 2 high-magnification micrographs showing CO2 pockets in appendix A6.
Line 258 – Replace “In” by “When”.
Re: revised as suggested.
Line 258 – Rephase “…melt mostly tended to aggregate.
Re: We revised this sentence to “The light phase at triple junctions is Ca-rich carbonate melt and the dark phase is Mg-rich carbonate melt. Figures A3(a) and (b) show the reaction of the sample with the alumina piston.” and moved this to the appendix.

Reviewer 2 Report
I am really pleasant to read an article of high quality like yours. It is one of the best articles which i have recently read. It is a great and interesting idea that has been suitably designed. In my opinion, all the methodologies used are excellent and very useful for this study. Furthermore, all the images, plots and diagrams given in the text are really helpful and clear for the readers.
I clearly suggest this manuscript to be published in the journal in the present form as it is an excellent one.
The article investigates the microstructure in a two-phase aggregate with supercritical CO2 as the fluid phase and examines the theory and model, as CO2 is different from basaltic melt in rheological properties. This article will be very interesting for a wide range of geologists such as petrologists, geotechnical geologists e.t.c.
The topic of this article is original.
The article is well organized/structured and written by the authors and hence it is clear and easy for the readers to read and to understand it. The conclusions have answered to the main objectives of the paper and they are also compact and useful for any reader.
Author Response
Thank you for your positive comments!!
Reviewer 3 Report
Review comments on “Influence of CO2 on the microstructure in sheared olivine aggregates” by Huihui Zhang et al. submitted to Minerals
This manuscript reports a study on influence of CO2 on the deformation microstructure of olivine aggregates. This is the first experimental study which examined effect of CO2 on crystallographic- and shape-preferred orientation of polycrystalline olivine. Samples deformed at 1 GPa and 1100°C show AG-type fabric which is consistent with basaltic-melt bearing samples in the previous study. The reported experimental data are new and valuable for understanding of influence of fluid phase on the CPO and SPO of peridotite. Therefore, I recommend publication of this manuscript in Minerals if the following points are adequately revised.
1.
The authors conducted a strain rate-stepping experiment (W2263) in which steady state stress was determined at four different strain rates. However, the authors do not determine stress exponent (n). I strongly suggest the authors to determine n value and add some discussion on deformation mechanism suggested from the n value, otherwise presence of this experiment is not meaningful. The stress exponent seems to be close to 2, and this is similar to those of sheared olivine + basaltic melt by Qi et al. (2018) (and is suggesting activity of dislocation-glide?).
2.
The sheared samples show CO2-enriched bands, and strain is probably localized in the bands. However, CPO and SPO in the sample were analyzed with no distinction of band and no-band regions. It is very curious whether there is difference in CPO and SPO between these two regions. I suggest the authors to investigate this difference.
3.
The authors concluded that the influence of CO2-fluid on the deformation microstructure of olivine aggregate is similar to that of basaltic melt although these two liquids have quite different physical properties (dihedral angle and viscosity). What does this mean? Why difference of dihedral angle is unimportant? I feel some description on this matter is needed in “4. Discussion”.
Minor comments
Line 223: Please give an explanation for “LVDT”.
Line 316: “grins” -> “grains
Author Response
The comments are noted in black, and the replies are noted in blue.
General comments:
This manuscript reports a study on influence of CO2 on the deformation microstructure of olivine aggregates. This is the first experimental study which examined effect of CO2 on crystallographic- and shape-preferred orientation of polycrystalline olivine. Samples deformed at 1 GPa and 1100°C show AG-type fabric which is consistent with basaltic-melt bearing samples in the previous study. The reported experimental data are new and valuable for understanding of influence of fluid phase on the CPO and SPO of peridotite. Therefore, I recommend publication of this manuscript in Minerals if the following points are adequately revised.
- The authors conducted a strain rate-stepping experiment (W2263) in which steady state stress was determined at four different strain rates. However, the authors do not determine stress exponent (n). I strongly suggest the authors to determine n value and add some discussion on deformation mechanism suggested from the n value, otherwise presence of this experiment is not meaningful. The stress exponent seems to be close to 2, and this is similar to those of sheared olivine + basaltic melt by Qi et al. (2018) (and is suggesting activity of dislocation-glide?).
Re: We obtained a stress exponents n = 1.8 of W2263, and have added this information in Lines 251-253, accordingly.
- The sheared samples show CO2-enriched bands, and strain is probably localized in the bands. However, CPO and SPO in the sample were analyzed with no distinction of band and no-band regions. It is very curious whether there is difference in CPO and SPO between these two regions. I suggest the authors to investigate this difference.
Re: We added Figure 6 (Lines 296-303) to show the difference of CPO in band and no-band regions. Figure 6 shows that the CPO is somewhat different in the band and no-band regions. We added a new subsection (3.5) to explain this in the revised manuscript in Lines 320-332.
The initial Figure 6 is changed to Figure 7 in the revision. In the Figure 7c and 7d, the crystallographic orientations and the rose diagrams were rotated to 90º such that the shear direction is the same in all of the pole figures.
- The authors concluded that the influence of CO2-fluid on the deformation microstructure of olivine aggregate is similar to that of basaltic melt although these two liquids have quite different physical properties (dihedral angle and viscosity). What does this mean? Why difference of dihedral angle is unimportant? I feel some description on this matter is needed in “4. Discussion”.
Re: We added a new subsection (4.1 The formation of CO2-enriched bands) to explain this in the revised manuscript in Lines 334-346.
Segregation of the fluid phase occurred during deformation of our samples, as illustrated in Figure 4. Fluid/melt-enriched bands formed in a similar manner to those reported by Holtzman et al. (2003) and King et al. (2010). Different from previous studies, in which basaltic melt is the fluid phase, in our samples, the fluid phase consists of carbonate melt and supercritical CO2. The dihedral angle of CO2 is ~80º-90º, too large to form an interconnected fluid network. However, the dihedral angle of carbonate melt is much smaller, ~ 30º (Hunter & Mckenzie, 1989), sufficient for the formation of an interconnected fluid network and thus the migration of the fluid phase. However, if only carbonate melt could migrate, CO2 should remain roughly uniform distribution. As illustrated in Figure 4, not many CO2 pores exist in the fluid-depleted regions, suggesting CO2 may migrate together with carbonate melt. Further investigations on such two-phase system are necessary to examine this interpretation (Bruhn et al. 2000).
References
- Holtzman, B.K.; Groebner, N.J.; Zimmerman, M.E.; Ginsberg, S.B.; Kohlstedt, D.L. Stress-driven melt segregation in partially molten rocks. Geochemistry, Geophysics, Geosystems 2003, 4, 8607, doi:10.1029/2001GC000258.
- King, D.S.H.; Zimmerman, M.E.; Kohlstedt, D.L. Stress-driven Melt Segregation in Partially Molten Olivine-rich Rocks Deformed in Torsion. Journal of Petrology 2010, 51, 21–42, doi:10.1093/petrology/egp062.
- Hunter, R.H.; & McKenzie, D. The equilibrium geometry of carbonate melts in rocks of mantle composition. Earth Planet. Sci. Lett, 1989, 92, 347–356.
- Bruhn, D.; Groebner, N. J.; & Kohlstedt, D. L. An interconnected network of core-forming melts produced by shear deformation. Nature, 2000, 403(6772), 883-886.
Minor comments:
Line 223: Please give an explanation for “LVDT”.
Re: revised as suggested.
Line 316: “grins” -> “grains.
Re: revised as suggested.
